# Electronic Transport Properties of Silicane Determined from First Principles

**DOI:** 10.3390/ma12182935

**Published:** 2019-09-11

**Authors:** Mohammad Mahdi Khatami, Gautam Gaddemane, Maarten L. Van de Put, Massimo V. Fischetti, Mohammad Kazem Moravvej-Farshi, Mahdi Pourfath, William G. Vandenberghe

**Affiliations:** 1Department of Material science and Engineering, The University of Texas at Dallas, Richardson, TX 75080, USA; MohammadMahdi.Khatami2@UTDallas.edu (M.M.K.); gautam.gaddemane@utdallas.edu (G.G.); maarten.vandeput@utdallas.edu (M.L.V.d.P.); max.fischetti@utdallas.edu (M.V.F.); 2Faculty of Electrical and Computer Engineering, Tarbiat Modares University, Tehran 1411713116, Iran; kazem.moravvej@gmail.com; 3School of Electrical and Computer Engineering, University of Tehran, Tehran 14395-515, Iran; pourfath@ut.ac.ir

**Keywords:** silicane, phonon scattering, mobility, Monte Carlo, DFT, DFPT

## Abstract

Silicane, a hydrogenated monolayer of hexagonal silicon, is a candidate material for future complementary metal-oxide-semiconductor technology. We determined the phonon-limited mobility and the velocity-field characteristics for electrons and holes in silicane from first principles, relying on density functional theory. Transport calculations were performed using a full-band Monte Carlo scheme. Scattering rates were determined from interpolated electron–phonon matrix elements determined from density functional perturbation theory. We found that the main source of scattering for electrons and holes was the ZA phonons. Different cut-off wavelengths ranging from 0.58 nm to 16 nm were used to study the possible suppression of the out-of-plane acoustic (ZA) phonons. The low-field mobility of electrons (holes) was obtained as 5 (10) cm^2^/(Vs) with a long wavelength ZA phonon cut-off of 16 nm. We showed that higher electron (hole) mobilities of 24 (101) cm^2^/(Vs) can be achieved with a cut-off wavelength of 4 nm, while completely suppressing ZA phonons results in an even higher electron (hole) mobility of 53 (109) cm^2^/(Vs). Velocity-field characteristics showed velocity saturation at 3 × 10^5^ V/cm, and negative differential mobility was observed at larger fields. The silicane mobility was competitive with other two-dimensional materials, such as transition-metal dichalcogenides or phosphorene, predicted using similar full-band Monte Carlo calculations. Therefore, silicon in its most extremely scaled form remains a competitive material for future nanoscale transistor technology, provided scattering with out-of-plane acoustic phonons could be suppressed.

## 1. Introduction

While scaling of complementary metal-oxide-semiconductor (CMOS) technology is facing serious issues such as mobility degradation, drain-induced barrier lowering, increasing tunneling current through the gate oxide, and source-to-drain tunneling, two-dimensional (2D) materials are under investigation as promising candidates to overcome these issues [1]. The high mobility of carriers in graphene [2] along with the possibility of wafer-scale fabrication [3,4,5,6] make graphene a very attractive material for electronic applications. However, any material to be used as a channel in a performant CMOS technology should exhibit not only a high carrier mobility but also a sufficiently large bandgap (more than 0.4 eV for I_on_/I_off_ of ~10^4^) to realize a sufficiently low off-current [7,8]. Therefore, despite having high mobility, associated with its Dirac dispersion, the use of graphene as a channel material is limited because of the absence of a bandgap.

On the other side, silicene, the silicon analog of graphene, can be considered to be the limit of an extremely scaled silicon CMOS technology and may be easier to integrate than graphene. Silicene has attracted a significant amount of attention, especially after the first experimental demonstration of a metal-oxide-semiconductor field-effect transistor (MOSFET) with a silicene channel [9]. While silicene shows many resemblances to graphene, one key difference is the presence of buckling of the crystal, breaking the horizontal mirror symmetry. For potential CMOS application, silicene suffers from an extremely low phonon-limited mobility, which is almost zero [10], and its negligible bandgap (~2 meV [11]), resulting in poor performance in both the on- and off-state of a silicene device.

The poor mobility stems from the strong scattering with the out-of-plane acoustic phonon (ZA) modes, which is permitted at first order by the broken horizontal mirror symmetry. As was shown in the detailed theoretical study by Fischetti and Vandenberghe [10], the parabolic phonon dispersion relation, combined with the Dirac dispersion, quenches the theoretical phonon-limited mobility in free-standing silicene to values as low as 0.01 cm^2^/(Vs). Note that some more primitive calculations in previous papers did not effectively account for the ZA phonon coupling and predicted much higher values [12,13,14]. Silicene–substrate interaction will somewhat alleviate mobility depression by suppressing ZA phonon scattering [15,16]. The effect of suppression of ZA phonon scattering can be crudely approximated by introducing a cut-off frequency. However, with realistic cut-offs, the carrier mobility in silicene is unlikely to reach a competitive value [10,17].

The absence of a bandgap must be alleviated by inducing a larger bandgap. Several methods of inducing a bandgap have been investigated, including strain, perpendicular electric fields, and hydrogenation. The use of tensile and compressive strain [18,19,20] has been shown to be ineffective and applying a perpendicular electric field [21,22] requires an excessively high electric field when applied externally [23]. However, studies based on density functional theory (DFT) have shown that hydrogenated silicene, called silicane, has a large bandgap, exceeding 2 eV [24,25,26,27,28,29].

The silicane crystal structure can come in two major configurations, a chair-like and a boat-like configuration. Stability studies based on total-energy calculations have shown that the chair-like configuration is more stable [27,28], while Houssa et al. [29] have shown both configurations have a similar stability. Unfortunately, silicane, similar to silicene, does not have horizontal mirror symmetry, (as can be seen in Figure 1) and will also suffer from scattering with out-of-plane acoustic phonons. However, the gapped silicane band structure does not display a Dirac cone and will have less pronounced backscattering compared to the out-of-plane acoustic phonon scattering in silicene [10], offering the prospect of a better mobility.

The silicane band structure has been studied previously but a precise theoretical study of its carrier mobility has not been conducted. The only work on the silicane mobility that we are aware of was performed by Restrepo et al. [24], who reported a mobility of 464 cm^2^/(Vs) for electrons, based on the linearized Boltzmann transport equation. However, this study did not provide detailed information on the contribution of the different phonon branches, nor did it discuss how the parabolic nature of the out-of-plane phonon modes was considered. A detailed and precise treatment is very important and a lack of an accurate treatment has led to an overestimation for the mobility in studies of several other materials, as discussed by Gaddemane et al. [17]. Moreover, to the best of our knowledge, no work has been done to estimate the hole mobility in silicane.

In this paper, we studied both the electron and hole mobility in silicane in its chair-like structure. In Section 3.1, the silicane electronic is determined using the DFT framework. In Section 3.2, the interaction matrix elements between the carriers and the phonons are calculated and, in Section 3.3, the matrix elements are used to calculate the phonon scattering rates. In Section 3.4 the Boltzmann transport equation (BTE) is solved using a full-band Monte Carlo scheme to obtain electron and hole mobility. Finally, in Section 3.5, the velocity-field characteristics are investigated, and the average energy of carriers and their distribution is used to better explain the observed velocity-field behavior. Comparing the calculated silicane mobility with that of other gapped 2D materials, we conclude that silicane, i.e., a hydrogen terminated mono-layer of silicon, is a competitive material candidate for future electronic devices, provided scattering with out-of-plane phonons can be sufficiently suppressed.

## 2. Computational Methodology and Silicane Structure

### 2.1. Methodology

The DFT calculations were performed using the QUANTUM ESPRESSO package (QE) [30,31] using norm-conserving pseudopotentials [32]. We provide the input files in the Appendix A. A zone-centered Monkhorst-Pack k grid [33] of 24 × 24 × 1 was used to sample the reciprocal lattice in the self-consistent charge density calculations. A maximum kinetic energy cut-off of 50 Ry was used for the plane-wave basis set for the psesudo-wavefunctions, while 200 Ry was used for the charge density and the potentials. Our calculations used H 1s^1^ and Si 3s^2^3p^2^ electrons as the valence electrons. Calculations were performed within the generalized gradient approximation (GGA) of the Perdew–Burke–Ernzerhof (PBE) [34] form. The structural optimization was performed with a force convergence-threshold of 2.5 × 10^−3^ eV/Å and a total energy threshold of 3 × 10^−5^ eV. A vertical vacuum space of 20 Å was inserted to minimize the artificial out-of-plane interaction between adjacent silicane layers.

Phonon calculations were performed using the DFPT method [35], as implemented in QE, with a 12 × 12 × 1 q-grid sampling of the first Brillouin zone (FBZ). The electron (hole) phonon matrix element for a transition of an electron (hole) from an initial state band *i* with wave vector **k** to a final state in band *j* with wave vector of **k** + **q**, was determined by [36]:(1)$$g_{ij}^{\nu }\left(k,\;q\right)=\sqrt{\frac{\hbar }{2M_{\mathrm{cell}}\omega_{\nu ,q}}}\left\langle \psi_{j,\;k+q}|\partial V_{q,\;\nu }^{\mathrm{SCF}}|\psi_{i,\;k}\right\rangle$$
where *ħ* and *M*_cell_ are the reduced Planck’s constant and the mass of the unit cell, respectively. $\partial V_{q,\;\nu }^{\mathrm{SCF}}$ corresponds to the change in the Kohn–Sham potential associated with the phonon mode ν with frequency $\omega_{\nu ,q}$ and wave vector **q**. In contrast to the finite displacement method [37,38], density functional perturbation theory calculations do not require the use of a supercell to limit the interaction among neighboring cells. Silicane, a non-polar material, will also be minimally affected by the spurious interaction among adjacent supercells that was reported by Sohier, et. al. [39] in the computation of electron–phonon matrix elements for polar 2D materials in the long wavelength regime (q→0).

The full-band Monte Carlo calculations required the electronic eigenvalues, phonon eigenfrequencies, and the electron–phonon interaction matrix elements on fine electron (**k**) and phonon (**q**) grids [14]. Electronic eigenvalues were extracted on a very fine mesh of 150 × 150 × 1 to provide a good resolution of carriers’ velocity for mobility calculations. However, rather than increasing the DFPT grid resolution, which is computationally expensive, we used an interpolation technique using maximally localized Wannier functions [38,40], as implemented by the Electron-phonon Wannier (EPW) code [41]. Specifically, we interpolated the phonon energies and electron–phonon matrix elements to a fine 50 × 50 × 1 grid using 10 Wannier functions.

The scattering rates of electrons and holes were then calculated following Fermi golden rule and accounting for a phonon bath in equilibrium at room temperature (*T* = 300 K):(2)$$\frac{1}{\tau_{i,k}^{\nu }}=\frac{2\pi }{\hbar }\underset{q,j}{\sum }{\left|g_{qk}^{\left(ij\right)\nu }\right|}^{2}\left[N_{\nu ,q}\delta \left(E_{j,k+q}-\hbar \omega_{\nu ,q}-E_{i,k}\right)+\left(N_{\nu ,q}+1\right)\delta \left(E_{j,k-q}+\hbar \omega_{\nu ,q}-E_{i,k}\right)\right]$$
where the first and second terms inside the square bracket on the right side of the equation represent absorption and emission processes, respectively. $N_{\nu ,q}$ is the phonon occupation number which is determined by the Bose–Einstein distribution:(3)$$N_{\nu ,q}={\left(e^{\frac{\hbar \omega_{\nu ,q}}{k_{B}T}}+1\right)}^{-1}$$
where *k*_B_ is the Boltzmann constant and *T* the temperature.

To obtain the transport properties of electrons and holes, the BTE is solved stochastically using the full-band Monte Carlo method previously used in Reference [17] to study electronic transport in phosphorene. The Monte Carlo calculations were performed for intrinsic silicane in the absence of an electric field to estimate the mobility and in the presence of various uniform electric fields to obtain the high-field behavior. The mobility was estimated from the diffusion constant to reduce the statistical fluctuations associated with the carrier velocity at low fields [42], while the velocity-field characteristics were determined directly from the average carrier velocity in steady state. In our Monte Carlo study, we considered an ensemble of 500 carriers that evolved in time using a time-step of 0.1 fs. By inspecting the average velocity of electrons and holes, we determined the transient period for low electric fields to last 30 ps. Calculations were then continued until a total of 100 ps to gather sufficient statistical data at steady state. All mobilities and velocities shown were obtained by averaging over the last 70 ps of the simulation.

### 2.2. Structure

The chair-like atomic configuration of silicane is shown in Figure 1. Silicane has a buckled honeycomb structure composed of Si atoms accompanied by H atoms alternatingly on the top and bottom. Hydrogenation of the Si atoms changes the band hybridization of Si atoms from a sp^2^–sp^3^ hybrid in silicene to pure sp^3^ in silicane, which makes the Si hybridization strongly resemble the one in bulk Si. The extracted structural and electronic parameters of silicane are listed in Table 1, which are in good agreement with earlier works [25,26]. The lattice constant of silicane was obtained as 3.88 Å which is slightly larger than the lattice constant of silicene (3.87 Å) and also the similar distance between Si atoms in the <111> direction of Si bulk (3.867 Å). The buckling height was 0.715 Å, compared to only ~0.44 Å [43] in silicene and close to the buckling height along the <111> direction of bulk Si (0.72 Å).

## 3. Results and Discussions

### 3.1. Electronic Band Structure and Phonon Dispersion

Figure 2 shows the calculated band structure of silicane. A Γ–M indirect gap measuring 2.19 eV was obtained which was 0.145 eV lower than its direct bandgap at Γ. To get a better overview of the energy landscape, the first conduction band and the light hole and heavy hole band energies are plotted as a function of the wave vector in the entire FBZ in Figure 2b–d. The conduction-band minimum and valence-band maximum were found at the M and Γ points, respectively. 

The ellipsoidal shape of the equi-energy contours of the conduction band implies an anisotropic effective mass of the electrons near the minimum of the band $\left(M_{x\left(y\right)}^{*}\propto {\left(\partial^{2}E/\partial k_{x\left(y\right)}^{2}\right)}^{-1}\right)$. The effective mass of electrons was ~27 times higher along the zigzag (*y*) direction (M→Γ) compared to the effective mass along the armchair (*x*) direction (M→K), as denoted in Table 1. The hole effective mass, on the other hand, is the same along the *x* and *y* directions because of the rotational symmetry at Γ, the location of the valence band maximum.

Figure 3 illustrates the silicane phonon dispersion along high symmetry points of the FBZ. A total of 12 phonon branches, associated with the four atoms in the unit cell, are distinguished. All phonon energies were positive, as required for material stability. The ZA phonon had a parabolic relation near the Γ point ($\omega =bq^{2}$ where $b=2.47\times 10^{-7}\;m^{2}/s$), as is always the case in free-standing 2D materials [10,44]. The dispersion of the two remaining acoustic modes, LA and TA, was linear near the Γ point with sound velocities $v_{\mathrm{TA}}=5.5\times 10^{3}\;\mathrm{cm}/s$ and $v_{\mathrm{LA}}=8.9\times 10^{3}\;\mathrm{cm}/s$.

Table 2 indicates the phonon energies of the 12 phonon branches at the high symmetry points of the first Brillouin zone of silicane. Among the 12 phonon branches, the two phonons with very large energy (~262 meV) were associated with the movement of the hydrogen atoms and were expected to be of minimal importance for electronic transport.

### 3.2. Interaction Matrix Elements

In low-field calculations of electronic transport, only the first conduction band will participate in conduction due to the large energy difference among the minima of the first two conduction bands (~1.5 eV). On the other hand, both of the top valence bands (i.e., light hole and heavy hole bands), which are degenerate at Γ, will participate in hole conduction. To better understand the scattering mechanisms, Figure 4 plots the magnitude of the electron–phonon interaction matrix elements for scattering from an electron placed at the conduction band minimum, i.e., the M point, $k=\pi /a\;\left(0,\;\;2/\sqrt{3}\right)$, as a function of phonon wave vector, **q** of the final state in the same band. Also, a schematic of the inter-valley scattering mechanisms is depicted in Figure 5. Analysis of the matrix elements shows that the main scattering mechanism appears to be intra-valley scattering (**q** = 0) and the inter-valley scattering between similar valleys from M to M’ and M” $\left(q=\pi /a\left(\pm 1,\;\;\pm 1/\sqrt{3}\right)\right)$. The inter-valley scattering mechanism from M to Γ $\left(q\;\sim \;\pi /a\left(0,\;\;\pm 2/\sqrt{3}\right)\right)$ shows smaller matrix elements and will not significantly contribute to the scattering rate.

Figure 6 shows the hole–phonon interaction matrix elements as a function of **q** over the FBZ for the holes placed at the valence band maximum (**k** = 0). It should be noted that due to the degeneracy of the LH and HH bands at the Γ point, the band index of the initial state cannot be unambiguously determined and distinguishing inter-band from intra-band scattering is not meaningful for holes located at the valence band maximum. Therefore, for the holes at valence band maximum, an averaged *g* is defined as $g_{i}^{\nu }\left(k=0,\;q\right)=\sqrt{{\left(g_{LH,i}^{\nu }\left(k=0,\;q\right)\right)}^{2}+{\left(g_{HH,i}^{\nu }\left(k=0,\;q\right)\right)}^{2}}$ where *i* is the index of the final state, either in the 1st hole band (LH) or the 2nd hole band (HH). Figure 6 shows that there will be significant intra-valley scattering at Γ as well as inter-valley Γ–M and Γ–K scattering. However, at low fields the intra-valley scattering in the Γ point will dominate since Γ–M and Γ–K scattering requires much higher hole energy, which is not available at low fields.

### 3.3. Scattering Rates

Like in all free-standing 2D materials, the scattering rates in silicane diverge as mandated by the Mermin–Wagner theorem [10,45]. The divergence stems from the diverging population of the out-of-plane phonons (ZA) which have a parabolic dispersion at long wavelengths. To maintain 2D material structural stability, there will be physical mechanisms that avoid the phonon population divergence. For example, Geim and Novoselov [3] proposed that finite grain sizes and wrinkles could provide such a mechanism but this mechanism of avoiding the divergence is not expected to dramatically affect mobility calculations. Another mechanism that will avoid the divergence is related to the interaction of the 2D material with a substrate. In this case, the hybridization of the ZA phonons with the Rayleigh waves of the substrate result in a gapped and broadened dispersion of the ZA phonons as studied by Amorim and Guinea [46] and also Ong and Pop [47].

To account for possible different mechanisms avoiding ZA phonon divergence, we adopted a crude simplification and calculated scattering rates assuming complete suppression of the ZA phonons with wavelengths higher than a specific cut-off wavelength. The smallest cut-off wavelengths correspond to a very optimistic scenario where all scattering with ZA phonons is suppressed. A more precise treatment of the ZA phonon/substrate interaction, beyond the scope of our current study, is required to determine how close silicane can come to this limit of “ignoring” scattering due to the ZA phonons.

Figure 7 shows the total scattering rates of electrons and holes in silicane, for each of the phonon branches, at room temperature (T = 300 K) for the cut-off wavelength of 16 nm for the ZA phonons. The magnitude of the wavevector associated with this wavelength coincides with the magnitude of the first EPW phonon grid wavevector after Γ: $q=4\pi /\sqrt{3}a\times 1/50$. As expected, Figure 7 shows that the acoustic modes, especially the ZA phonon modes, have the largest scattering rates for both electrons and holes. The scattering rate with ZA phonons was almost two orders of magnitude higher than the others and its large magnitude stemmed from the parabolic ZA phonon dispersion. Specifically, the parabolic dispersion, resulting in small phonon energies near the Γ point yielded huge scattering rates, in agreement with Equations (2) and (3). Higher cut-off wavelengths resulted in higher ZA phonon scattering rates while lower ZA phonon scattering rates were observed for lower cut-offs.

The step-like increases in the optical phonon scattering rates of electrons and holes observed in Figure 7 can be attributed to the onset of phonon emission when the electron energy exceeded the phonon energy of each optical branch. An additional sharp step in the electron–phonon scattering rates, caused by the onset of inter-valley scattering has been reported in MoS_2_ and WS_2_ [14,48] phosphorene. In silicane, such a step was not observed thanks to the smaller inter-valley M–Γ scattering matrix elements, when compared to strong intra-valley scattering in the M valley and inter-valley scattering between the M, M’, and M” valleys.

### 3.4. Mobility Calculations

Performing diffusion-constant Monte Carlo calculations, we obtained the low-field mobility values for electrons and holes for four different cut-off wavelengths of 16, 1, 4, and 0.5 nm, as shown in Table 3. The 0.58 nm cut-off wavelength represents a wave vector magnitude equal to the first Brillouin zone edge (q=4π/3a where *a* is the silicane lattice constant) and, thus, no scattering at all due to the ZA phonons which were taken into account. In this case, the highest mobility for electrons was 53 cm^2^/(Vs) and for the holes, 109 cm^2^/(Vs). Mobility values for the armchair and zigzag direction were identical because of the 120° rotational symmetry in silicane. The calculated electron mobility was significantly lower than the 1290 cm^2^/Vs [49] found in silicene when neglecting the scattering with the ZA phonons. 

Inter-valley transitions from M to M’ and M” require phonons with a wave vector of magnitude $q\approx 2\pi /\sqrt{3}a$, or equivalently, a wavelength of around 0.67 nm. Thus, with a cut-off wavelength of 1 nm, M–M’–M’’ scattering processes were allowed for electrons and a lower electron mobility was obtained. No change in hole mobility was observed since there was only intra-valley scattering around the Γ point. For a 4 nm wavelength, intra-valley scattering processes were allowed for electrons which resulted in a lower mobility. This reduction in mobility of both electrons and holes will continue with increasing cut-off wavelengths, as shown up to 16 nm. 

### 3.5. Velocity Field Calculations

Next, we calculated the velocity of electrons and holes under the application of an electric field and the results are presented in Figure 8. The slope of the velocity-field characteristics (Figure 8a,b) at low fields for both electrons and holes matched well with the mobility obtained from the calculated diffusion constant. At higher fields, the velocity-field characteristics were no longer linear. The deterioration of the slope for electrons and holes can be understood from Figure 8c,d, which shows that average energy of the electrons and holes increases rapidly beyond a critical applied electric field of about 2 × 10^5^ V/cm. As the average energy increases significantly, electrons gained enough energy to scatter to the Γ valley, decreasing the mobility at high fields.

Figure 9 shows the electron and hole distributions in k-space at the final step of our Monte Carlo run to illustrate the impact of the electric field. Figure 9a shows that, at low fields, electrons were confined to the conduction-band maximum at the M point. However, at higher electric fields, electrons were no longer confined around the M point and occupied states with reduced velocity. This resulted in velocity saturation at a field of 3 × 10^5^ V/cm and a negative differential mobility for fields exceeding 4 × 10^5^ V/cm.

For holes, Figure 9b shows the occupation near the valence band maximum at the Γ-point at low fields. Increasing the electric field resulted in the scattering of holes to the M point where the valence bands were almost flat. The flat-band regions featured extremely low velocities (v∝∂E/∂k) which in turn resulted in velocity-field saturation at a field measuring 3 × 10^5^ V/cm and finally into negative differential mobility for fields exceeding 4 × 10^5^ V/cm. No Γ–K scattering of holes was observed up to fields of 5 × 10^5^ V/cm since scattering to the K point required an even higher hole energy than available at these electric fields.

### 3.6. Discussion

To assess the prospect of silicane for future electronic devices, we compared the computed silicane mobility with the mobility of other candidate semiconductor device channel materials. Phosphorene electron (hole) mobility was predicted as low as 10 (3) cm^2^/Vs and 21 (19) cm^2^/Vs in the zigzag and armchair direction, respectively [17]. Electron (hole) mobilities of MoSe_2_ were predicted as 25 (90) cm^2^/(Vs) [48]. Monolayer MoS_2_ had a higher predicted electron mobility of 130 cm^2^/(Vs) [14] and 150 cm^2^/(Vs) [50] and hole mobility of 270 cm^2^/(Vs) [48] and might be a better candidate for electronic devices. An electron mobility of 410 cm^2^/(Vs) was also reported in Reference [51], although this relatively higher mobility might come from their assumption that electron–phonon matrix elements are independent of the initial electron wave vector, reducing the accuracy of their model. Calculated electron (hole) mobilities for the transition-metal dichalcogenides (TMDs) WSe_2_ and WS_2_ were reported as 30 (270) cm^2^/Vs and 320 (540) cm^2^/Vs [48], respectively.

On the other hand, electron (hole) mobility of bulk Si is known to be 1450 (450) cm^2^/Vs [52]. However, the mobility of bulk Si is known to strongly reduce with decreasing Si layer thickness [53], e.g., in 1.5 nm Si layers, phonon-limited electron mobility drops to less than 550 cm^2^/Vs and, when accounting for surface roughness scattering, further reduces to 300 cm^2^/Vs under a transverse field of 10^4^ V/cm. An even more dramatic reduction in silicon mobility is expected when Si layer thickness becomes even smaller. At the atomic scale, silicene can be viewed as one possible extreme limit of silicon scaling, but as mentioned in the introduction, the issue with the vanishing bandgap can be addressed by hydrogenation which results in silicane. The calculated carrier mobility of silicane provides a fairly good estimate of the upper limit of extremely scaled Si.

Finally, it should be noted that PBE underestimates the electronic bandgap and effective masses. More advanced DFT approaches addressing these issues, such as hybrid functionals, are not currently implemented in EPW. For example, Poncé et al. [54] showed that the electron (hole) mobility in silicon will change by selecting different exchange correlation functionals or by including spin-orbit coupling. Therefore, despite our efforts to obtain maximal accuracy, some uncertainty will remain concerning the error of our final results. This error is in addition to all physical effects introduced by unaccounted physical mechanisms (e.g., defects, doping, substrate effect on band energies, and also additional possible scattering processes due to the substrate).

## 4. Conclusions

The silicane bandgap was determined to be 2.19 eV with a valence band maximum and a conduction band minimum at the Γ and M points, respectively. Phonon calculations, including phonon energies, and the interaction of electrons and holes with phonons were performed using density functional perturbation theory. We calculated scattering rates for both electrons and holes, and showed that the acoustic branches had the largest scattering rates. In particular, the ZA phonon branch, with its parabolic phonon dispersion near the Γ point, had a scattering rate exceeding all other scattering rates by at least two orders of magnitude. Different cut-off wavelengths ranging from 0.58 nm to 16 nm were used to investigate the upper limit of mobility when ZA phonon scattering could be suppressed at different levels.

We determined the silicane mobility and velocity-field characteristics using the Monte Carlo method. The electron and hole mobilities were found to be 5 cm^2^/(Vs) and 10 cm^2^/(Vs) for a cut-off wavelength of 16 nm. We showed that providing a cut-off wavelength of 4 nm increased electron mobility to 24 cm^2^/(Vs) and hole mobility to 101 cm^2^/(Vs), while complete supersession of ZA resulted in an even higher mobility of 53 cm^2^/(Vs) for electrons and 109 cm^2^/(Vs) for holes.

Finally, the impact of the electric field on electron and hole transport was studied, showing that the velocity starts to saturate at a field of 3 × 10^5^ V/cm and features negative differential mobility for fields exceeding 4 × 10^5^ V/cm, for both electrons and holes. This phenomenon was consistent with the average energy and the distribution of the Monte Carlo ensemble.

The mobility values we obtained for silicane are competitive with computed values found in the literature for other 2D materials such as phosphorene and some of the TMDs. Silicane, i.e., hydrogen passivated <111> silicon slab scaled to its monolayer limit, can be thought of as the ultimate limit of silicon scaling. Given the competitive mobilities, we conclude that even at the most extreme scaling, silicon can still be considered as a promising material for future CMOS technology.

## Figures and Tables

**Figure 1 materials-12-02935-f001:**
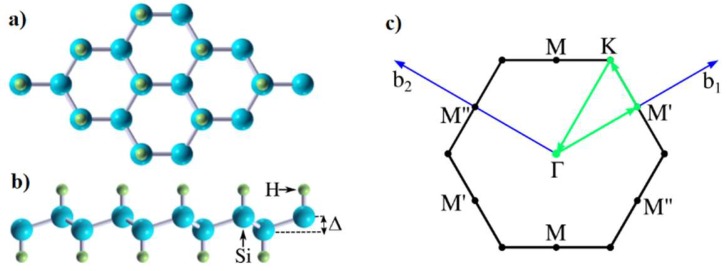
(**a**) Top view and (**b**) cross-view of the silicane structure; (**c**) reciprocal lattice of silicane in which b_1_ and b_2_ represent reciprocal lattice vectors and Γ, K, and M are the high symmetry points of the first Brillouin zone (FBZ).

**Figure 2 materials-12-02935-f002:**
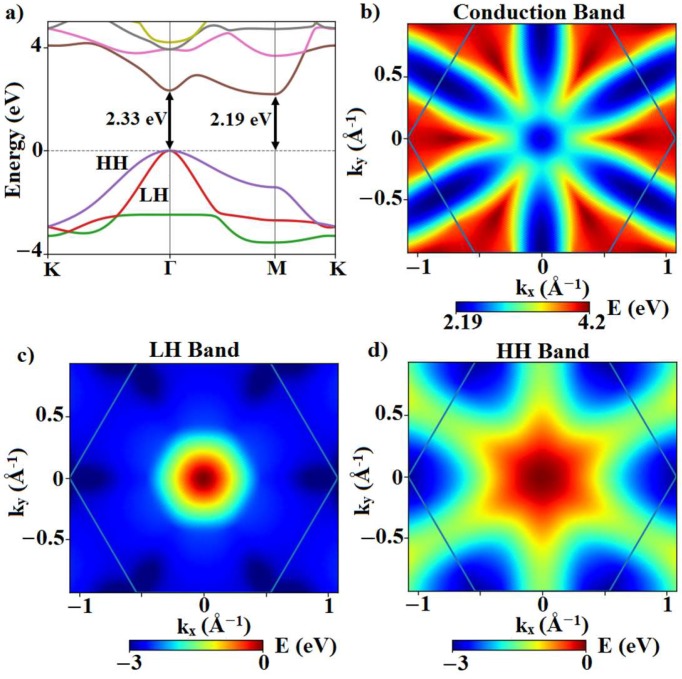
(**a**) The calculated electronic band structure of silicane along the high symmetry directions of the FBZ. (**b**–**d**) Respectively, conduction, light hole, and heavy holes valence band energies of silicane in the FBZ.

**Figure 3 materials-12-02935-f003:**
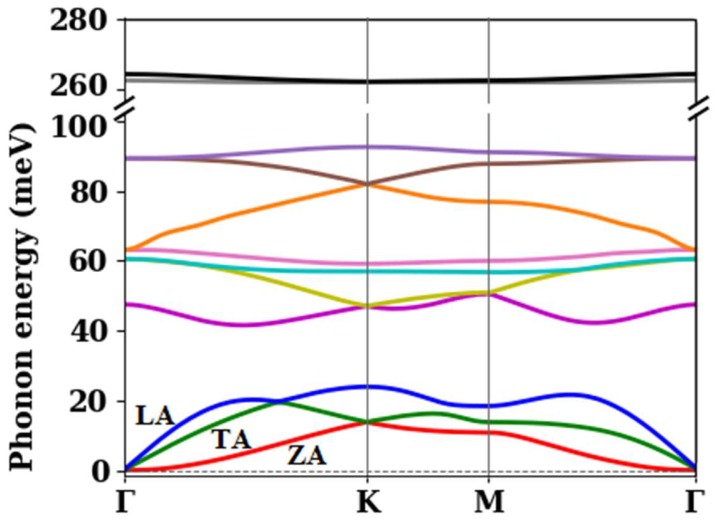
The calculated phonon dispersion of silicane along high symmetry directions of the FBZ. Note the parabolic dispersion of the out-of-plane phonon mode near the Γ point. The vertical axis was broken to capture the two high-energy phonon branches which were associated with the movement of the hydrogen atoms.

**Figure 4 materials-12-02935-f004:**
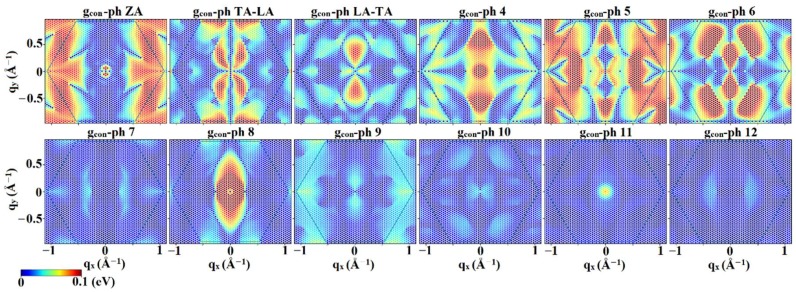
Electron–phonon interaction matrix elements in units of eV for initial **k** of the conduction band minimum (M valley) for different phonon branches. It can be observed that the dominant scattering mechanisms are the intra-valley and inter-valley (M→M’, M”) mechanisms. Other elements representing M→Γ scattering and also scattering to higher energy states will become significant for higher electric fields.

**Figure 5 materials-12-02935-f005:**
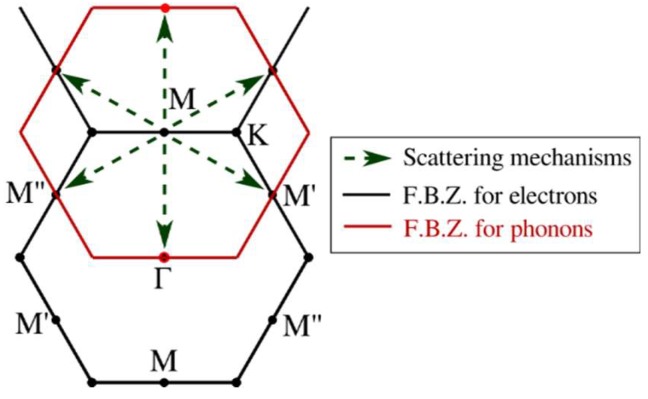
Schematic of possible inter-valley scattering mechanisms for electrons placed at conduction band minimum (M point).

**Figure 6 materials-12-02935-f006:**
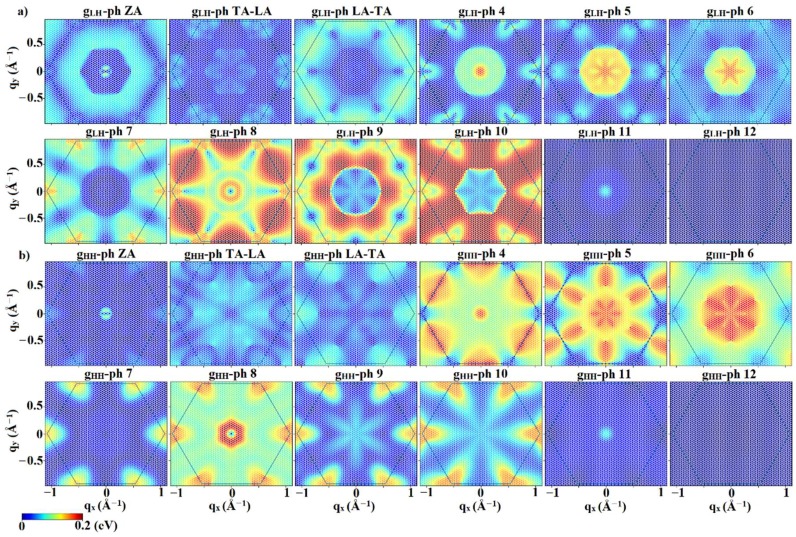
Hole–phonon interaction matrix elements in units of eV for (**a**) light hole valence band (lower one of the two top valence bands) and (**b**) heavy hole valence band for different phonon branches. The hole initial state **k** was assumed as the valence band maximum (Γ point).

**Figure 7 materials-12-02935-f007:**
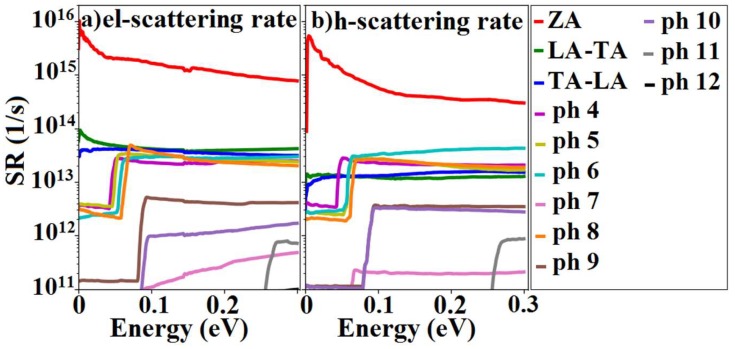
The scattering rates of (**a**) electrons and (**b**) holes in silicane which were calculated at room temperature with a cut-off wavelength of 16 nm. The zero-energy in electron/hole scattering rates shows conduction band minimum and valence band maximum. Note that the 12th phonon branch scattering rates for electrons and holes were less than 10^11^ s^−1^ and, thus, almost non-visible in these plots.

**Figure 8 materials-12-02935-f008:**
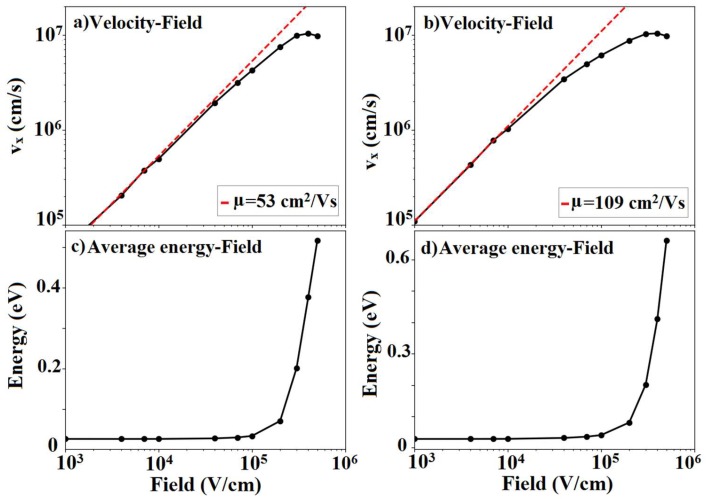
Velocity-field characteristics of (**a**) electrons and (**b**) holes. The average energy-field of (**c**) electrons and (**d**) holes in silicane calculated using the Monte Carlo method at room temperature (T = 300 K). Velocity saturation was observed at 3 × 10^5^ V/cm and a negative slope of velocity-field was observed for fields exceeding 4 × 10^5^ V/cm.

**Figure 9 materials-12-02935-f009:**
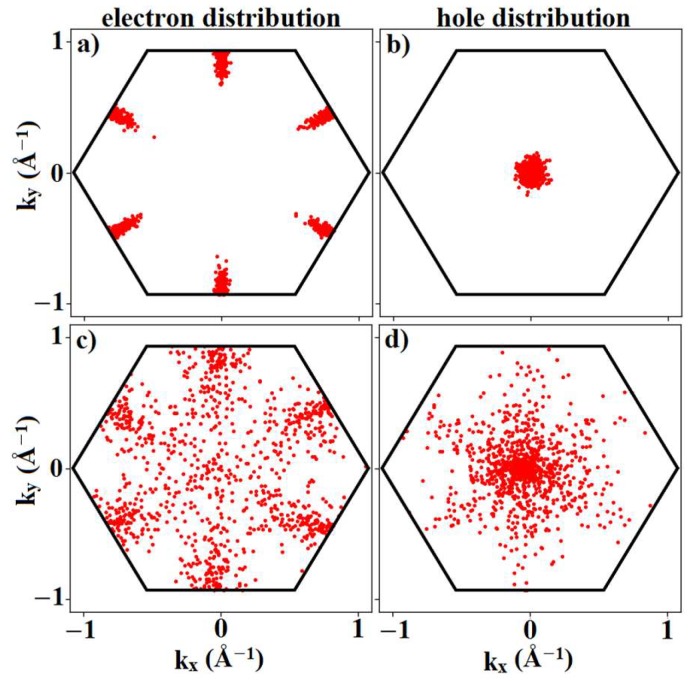
Distribution of (**a**,**c**) electrons and (**b**,**d**) holes in (**a**,**b**) low electric fields (E_field_ = 10^3^ V/cm) and (**c**,**d**) high electric fields (E_field_ = 5 × 10^5^ V/cm) in the FBZ of silicane at room temperature (T = 300 K). It can be seen at low electric fields that electrons (holes) were confined at the M (Γ) valley. Increasing electric fields resulted in the scattering of electrons from the M to the Γ valley and also resulted in the occupation of higher energy states. For holes, increasing the electric fields would increase occupation towards the M point.

**Table 1 materials-12-02935-t001:** Silicane characteristics.

Parameter	This Work	[26]	[25]
Lattice constant (Å)	3.887	3.84	3.889
Si–Si bond (Å)	2.44	2.34	2.358
Si–H bond (Å)	1.502	1.51	1.501
Buckling height Δ (Å)	0.715	0.74	0.719
m_e,x_ (m_0_)	0.13	0.12	0.123
m_e,y_ (m_0_)	3.5	3.79	3.23
m_lh,x_ (m_0_)	0.12	0.13	0.151
m_lh,y_ (m_0_)	0.12	**-**	0.142
m_hh,x_ (m_0_)	0.57	0.52	0.573
m_hh,y_ (m_0_)	0.58	**-**	0.603
Indirect gap (eV)	2.19	-	2.19
Direct gap (eV)	2.335	-	2.33

**Table 2 materials-12-02935-t002:** Silicane phonon energies (meV) at Γ, K, and M.

Phonon Branch Number	Γ	K	M
1	0	13.8	10.8
2	0	13.8	13.8
3	0	24.0	18.4
4	47.5	47.0	50.5
5	60.5	47.0	50.7
6	60.5	57.0	56.6
7	63.1	59.0	60.0
8	63.1	81.9	76.8
9	89.3	81.9	87.7
10	89.3	92.5	91.0
11	262.5	262.1	262.0
12	264.3	262.1	262.4

**Table 3 materials-12-02935-t003:** Phonon-limited electron and hole mobilities in silicane at room temperature (T = 300 K) for different cut-off wavelengths.

Cut-Off Wavelength (nm)	$\mu_{e}\;({\mathrm{cm}}^{2}V^{-1}s^{-1})$	$\mu_{h}\;({\mathrm{cm}}^{2}V^{-1}s^{-1})$
0.58	53	109
1	31	109
4	24	101
16	5	10

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
