# Peer review of "Electronic Transport Properties of Silicane Determined from First Principles"

_materials, 2019, doi:10.3390/ma12182935_

Round 1

Reviewer 1 Report

In this manuscript, transport properties of silicon based two-dimensional material, silicane, was investigated based on the first-principles and full-band Monte Carlo calculations. First, structural, electronic and phonon properties were obtained from the first-principles calculations. Then the mobility of electrons and holes as well as velocity field characteristics were obtained using full-band Monte Carlo calculations. These results were compared with other two-dimensional materials and the authors concluded that silicon, in its monolayer limit as a form of silicane, can still play a major role in future CMOS technologies.   The work stands out in its careful planning, execution, and comprehensiveness (many of the references have incomplete information, which must be fixed, though). I support the publication of the present manuscript in this journal.

Author Response

We thank the reviewer for his supportive comments. We apologize for the references and have completed the information in the reference section.

Reviewer 2 Report

The problem considered in the paper is very well known. Due to Mermin-Wagner theorem and its consequences it is not possible to achieve a satisfactory value of mobility of charges in the two-dimensional materials with broken horizontal mirror symmetry. The mobility is significantly lowered, because of electron and hole scattering on ZA phonons.

In the paper the Authors conducted the DFPT and MC calculations for silicane (fully hydrogenated siliecene) and obtain an easily predictable result - low mobility of charges, caused by the scattering of electrons and holes on ZA phonons. A possible solution of this problem is to suppress the ZA phonons and the Authors confirm this solution using different values of cut-off wavelength. This solution was already used for other materials including silicene, thus I cannot consider this as a new method. Moreover, this is a solution for calculations. In experiment it would require using a proper substrate, but the Authors do not consider this problem.

Moreover, the results shown by the Authors are not promising. Better values of mobility were obtained for other materials e.g. MoS2. 

I appreciate the Author's effort and neatness, but I am afraid that the paper shows no new insight into the problem of improving mobility of charges.

Author Response

The reviewer raises several important considerations. The paper unfortunately does not present an extraordinary mobility and some other materials, e.g. MoS2, are predicted to have somewhat higher mobility compared to the values we compute for silicane. However, without publication of our work, the community will not be aware what silicane’s mobility is and how competitive/uncompetitive it is. Another consideration we ask the reviewer to make is that if only “high mobility” values can be published, this will strongly skew the publication towards primitive, inaccurate calculations that overestimate mobility.

Regarding the “easily predictable result” we want to make following rebuttal. While it is relatively easy for an expert in the field, familiar with our previous work, to know that scattering with ZA phonons will result in a low mobility, many in community remain unaware of the importance of scattering with ZA phonons. Also, it was not predictable for us authors, who can be considered experts in the field, to predict what mobility silicane would boast both in the presence and in the absence of ZA phonon scattering. Any mobility ranging from 1 cm2/(Vs) to 1,000 cm2/(Vs) should be considered a reasonable guess prior to the publication of our results. With the publication of our study, the community will have a much better idea of what mobility can be expected.

We make no claims of a new methodology, but rather just a careful application of existing methodology to study the properties of silicane. Nor do we consider how a particular cut-off of ZA phonon scattering could be realized. These could be valuable investigations in their own right but go beyond the scope of our current paper. The analysis we present is entirely within the scope of the journal where many papers are published that do not use a new methodology.

Our paper does provide previously unavailable insight into the mobility of silicon in a monolayer form “silicane”, which represents an important data point for the community, and we point out the underlying mechanisms limiting the mobility. The investigation of a silicon-based material, compared to some alternative materials that could exhibit a larger mobility, remains valuable because of the natural compatibility with existing technology in today’s electronic industry. All these points make our manuscript a source of valuable information for the readers of “Materials”.

Reviewer 3 Report

The authors predicted multiple transport properties of silicane by using the first-principles approach and compared with the values present in the literature. I only have some queries.

1. Is there any reason behind choosing PBE functional? Why not BLYP or even try for HSE06 or other? Why the charge density cutoff is 200 Ry? 400 Ry are often used in such calculations in order to maintain higher accuracy.

2. The authors have discussed the validation of their computed results, however, it will be nice to also point out the drawback of first-principles calculation. I mean the requirement of computational resources or a particular error in other sets of properties and how one should approach to reduce it. This will also reflect the ability of first-principles calculations in predicting a certain set of properties accurately. 

3. Can authors also perform a set of simulation for computing mobility w.r.t the temperature? It would be interesting to see how it scales with temperature?

4. Is there an error associated with the hole or electron mobility values?

5. The authors can also provide a sample input file in their Supporting Information file in order to reproduce these results. 

Author Response

We thank the reviewer for his comments.

The EPW code only works with norm-conserving pseudo-potentials with GGA or LDA exchange-correlation models where GGA gives better results for the electronic properties motivating our choice for the latter. More advanced hybrid functionals are not implemented and could prove to be computationally prohibitive.
We have tested different kinetic energy cut-offs and observed that a kinetic energy cut of 50 Ry was necessary to get positive and parabolic phonon energies and also symmetric electron(hole)-phonon interaction matrix elements for phonon wave vectors close to the Γ point. We used higher cut-off energies (55 Ry and 60 Ry) during our investigation and no significant change was observed, while a significant decrease in the calculation speed was observed. For this reason we used the 50 Ry cut-off in our calculations. As EPW calculations require the charge density cut-off energy to be 4 times the kinetic energy cut-off (ecutrho=4*ecutwfc), we used a charge density cut-off of 200 Ry. We agree with the reviewer that the approach requires a significant effort. For our response to this point, please refer to the following text which is added to end of the “Discussion” section.
“Finally, it should be noted that PBE underestimates the electronic bandgap and effective masses. More advanced DFT approaches addressing these issues, such as hybrid functionals, are currently not implemented in EPW. For example, Poncé et al. have shown in Ref. [55] that the electron(hole) mobility in silicon will change by selecting different exchange/correlation functionals or by including spin-orbit coupling. Therefore, despite our efforts to obtain maximal accuracy, some uncertainty will remain concerning the error of our final results. This error is in addition to all physical the effects introduced by unaccounted physical mechanisms (e.g. defects, doping, substrate effect on band energies, and also additional possible scattering processes due to the substrate).” As the reviewer points out, we could compute the mobility as a function of temperature. To perform a study of temperature dependence, we would have to perform new simulations and analyze the results; this would require a significant effort. Such an analysis, while valuable, is not necessary for a meaningful publication. Nevertheless, we are considering a temperature-dependent study of the mobility in our future work. The results of the calculations do not have a conventional error bar. Energies are converged up to 2.7×10-5 eV but subsequent calculations do not have an established quantification of the error. We performed our calculations until we observed that the error in our calculation was “small” when changing cut-off energies, discretization, and different methods of determining mobility. This is a standard approach when performing DFT calculations. Based on Poncé’s results [55], the error may be up to 20%. We have included several of our input files in the Supporting Information. The Monte Carlo part of the code is not published but is relatively straightforward to implement.

Reviewer 4 Report

In my opinion, this study is interesting. The computational methodology is described in sufficient detail. The silicane bandgap was determined to be 2.19 eV. Phonon calculations, including phonon energies, and the interaction of electrons and holes with phonons were performed using DFPT method. The impact of the electric field on electron and hole transport has been studied. My several proposed corrections and questions are listed below for author consideration.

Line 47.

MOSFET

Every abbreviation must be explained.

Line 101:

…while 200 Ry is used for the charge density and the potentials.

Why such cut-off energy was chosen? Why not 150 of 250 Ry?

Please correct all references from Phys. Rev. Lett. And Phys. Rev. B. Some examples are listed below.

Line 382:

Neugebauer, P.; Orlita, M.; Faugeras, C.; Barra, A.L.; Potemski, M. How perfect can graphene be? Phys. 382 Rev. Lett. 2009, 103, 2–5.

Neugebauer, P.; Orlita, M.; Faugeras, C.; Barra, A.L.; Potemski, M. How perfect can graphene be? Phys. 382 Rev. Lett. 2009, 103, 136403.

Line 398.

Fischetti, M. V.; Vandenberghe, W.G. Mermin-Wagner theorem, flexural modes, and degraded carrier mobility in two-dimensional crystals with broken horizontal mirror symmetry. Phys. Rev. B 2016, 93, 1–13.

Fischetti, M. V.; Vandenberghe, W.G. Mermin-Wagner theorem, flexural modes, and degraded carrier mobility in two-dimensional crystals with broken horizontal mirror symmetry. Phys. Rev. B 2016, 93, 155413.

Line 395:

Available online: http://www.itrs.net/ Links/2009ITRS/Home2009.htm.

This link is not working. Maybe only for me.

Author Response

We thank the reviewer for noticing several important deficiencies in the manuscript. We have written out Metal-oxide-semiconductor field-effect transistor in full. We checked all references and corrected the problems identified by the reviewer.

Neugebauer, P.; Orlita, M.; Faugeras, C.; Barra, A.L.; Potemski, M. How perfect can graphene be? Phys. Rev. Lett. 2009, 103, 136403. Kim, K.S.; Zhao, Y.; Jang, H.; Lee, S.Y.; Kim, J.M.; Kim, K.S.; Ahn, J.-H.; Kim, P.; Choi, J.-Y.; Hong, B.H. Large-scale pattern growth of graphene films for stretchable transparent electrodes. Nature 2009, 457, 706–710. Lee, J.; Lee, E.K.; Joo, W.; Jang, Y.; Kim, B.; Lim, J.Y.; Choi, S.; Ahn, S.J.; Ahn, J.R.; Park, M.; et al. Wafer-Scale Growth of Single-Crystal Monolayer Graphene on Reusable Hydrogen-Terminated Germanium. Science. 2014, 344, 286–289. (2009) The International Technology Roadmap for Semiconductors (ITRS) Available online: http://www.itrs2.net. Fischetti, M. V.; Vandenberghe, W.G. Mermin-Wagner theorem, flexural modes, and degraded carrier mobility in two-dimensional crystals with broken horizontal mirror symmetry. Phys. Rev. B 2016, 93, 155413. Matthes, L.; Pulci, O.; Bechstedt, F. Massive Diracquasiparticles in the optical absorbance of graphene, silicene, germanene, and tinene. J. Phys.: Condens. Matter 2013, 25, 395305. Shao, Z.G.; Ye, X.S.; Yang, L.; Wang, C.L. First-principles calculation of intrinsic carrier mobility of silicene. J. Appl. Phys. 2013, 114, 093712. Yeoh, K.H.; Ong, D.S.; Ooi, C.H.R.; Yong, T.K.; Lim, S.K. Analytical band Monte Carlo analysis of electron transport in silicene. Semicond. Sci. Technol. 2016, 31, 065012. Li, X.; Mullen, J.T.; Jin, Z.; Borysenko, K.M.; Buongiorno Nardelli, M.; Kim, K.W. Intrinsic electrical transport properties of monolayer silicene and MoS2 from first principles. Phys. Rev. B 2013, 87, 115418. Seol, J.H.; Jo, I.; Moore, A.L.; Lindsay, L.; Aitken, Z.H.; Pettes, M.T.; Li, X.; Yao, Z.; Huang, R.; Broido, D.; et al. Two-Dimensional Phonon Transport in Supported Graphene. Science. 2010, 328, 213–216. Gaddemane, G.; Vandenberghe, W.G.; Van De Put, M.L.; Chen, S.; Tiwari, S.; Chen, E.; Fischetti, M. V. Theoretical studies of electronic transport in monolayer and bilayerphosphorene: A critical overview. Phys. Rev. B 2018, 98, 115416. Mohan, B.; Kumar, A.; Ahluwalia, P.K. Electronic and optical properties of silicene under uni-axial and bi-axial mechanical strains: A first principle study. Phys. E 2014, 61, 40–47. Qin, R.; Wang, C.H.; Zhu, W.; Zhang, Y. First-principles calculations of mechanical and electronic properties of silicene under strain. AIP Adv. 2012, 2, 022159. Drummond, N.D.; Zólyomi, V.; Fal’Ko, V.I. Electrically tunable band gap in silicene. Phys. Rev. B 2012, 85, 075423. Yan, J.A.; Gao, S.P.; Stein, R.; Coard, G. Tuning the electronic structure of silicene and germanene by biaxial strain and electric field. Phys. Rev. B 2015, 91, 245403. Restrepo, O.D.; Mishra, R.; Goldberger, J.E.; Windl, W. Tunable gaps and enhanced mobilities in strain-engineered silicane. J. Appl. Phys. 2014, 115, 033711. Zólyomi, V.; Wallbank, J.R.; Fal’ko, V.I. Silicane and germanane: Tight-binding and first-principles studies. 2D Mater. 2014, 1, 011005. Zhang, P.; Li, X.D.; Hu, C.H.; Wu, S.Q.; Zhu, Z.Z. First-principles studies of the hydrogenation effects in silicene sheets. Phys. Lett. A 2012, 376, 1230–1233. LewYanVoon, L.C.; Sandberg, E.; Aga, R.S.; Farajian, A.A. Hydrogen compounds of group-IV nanosheets. Appl. Phys. Lett. 2010, 97, 163114. Houssa, M.; Scalise, E.; Sankaran, K.; Pourtois, G.; Afanas’ev, V. V.; Stesmans, A. Electronic properties of hydrogenated silicene and germanene. Appl. Phys. Lett. 2011, 98, 223107. Vandenberghe, W.G.; Fischetti, M. V. Deformation potentials for band-to-band tunneling in silicon and germanium from first principles from first principles. Appl. Phys. Lett. 2015, 106, 013505. Elahi, M.; Pourfath, M. Ab initio effective deformation potentials of phosphorene and consistency checks. J. Phys.: Condens. Matter 2018, 30, 225701. Baroni, S.; Gironcoli, S. De; Corso, A.D.; Giannozzi, P. Phonons and related crystal properties from density-functional perturbation theory. Rev. Mod. Phys. 2001, 73, 515-562. Giannozzi, P.; Baroni, S.; Bonini, N.; Calandra, M.; Car, R.; Cavazzoni, C.; Ceresoli, D.; Chiarotti, G.L.; Cococcioni, M.; Dabo, I.; et al. QUANTUM ESPRESSO: A modular and open-source software project for quantum simulations of materials. J. Phys.: Condens. Matter 2009, 21, 395502. Giannozzi, P.; Andreussi, O.; Brumme, T.; Bunau, O.; BuongiornoNardelli, M.; Calandra, M.; Car, R.; Cavazzoni, C.; Ceresoli, D.; Cococcioni, M.; et al. Advanced capabilities for materials modelling with QUANTUM ESPRESSO. J. Phys.: Condens. Matter 2017, 29, 465901. Hamann, D.R. Optimized norm-conserving Vanderbilt pseudopotentials. Phys. Rev. B 2013, 88, 085117. Monkhorst, H.J.; Pack, J.D. Special points for Brillonin-zone integrations. Phys. Rev. B 1976, 13, 5188. Perdew, J.P.; Burke, K.; Ernzerhof, M. Generalized Gradient Approximation Made Simple. Phys. Rev. Lett. 1996, 77, 3865. Sohier, T.; Calandra, M.; Mauri, F. Two-dimensional Fröhlich interaction in transition-metal dichalcogenidemonolayers: Theoretical modeling and first-principles calculations. Phys. Rev. B 2016, 94, 085415. Marzari, N.; Vanderbilt, D. Maximally localized generalized Wannier functions for composite energy bands. Phys. Rev. B 1997, 56, 12847. Giustino, F.; Cohen, M.L.; Louie, S.G. Electron-phonon interaction using Wannier functions. Phys. Rev. B 2007, 76, 165108. Cahangirov, S.; Topsakal, M.; Akturk, E.; Sahin, H.; Ciraci, S. Two- and one-dimensional honeycomb structures of silicon and germanium. Phys. Rev. Lett. 2009, 102, 236804. Amorim, B.; Guinea, F. Flexural mode of graphene on a substrate. Phys. Rev. B 2013, 88, 115418. Ong, Z.Y.; Pop, E. Effect of substrate modes on thermal transport in supported graphene. Phys. Rev. B 2011, 84, 075471. Jin, Z.; Li, X.; Mullen, J.T.; Kim, K.W. Intrinsic transport properties of electrons and holes in monolayer transition-metal dichalcogenides. Phys. Rev. B 2014, 90, 045422. Li, W. Electrical transport limited by electron-phonon coupling from Boltzmann transport equation: An ab initio study of Si, Al, and MoS2. Phys. Rev. B 2015, 92, 075405. Kaasbjerg, K.; Thygesen, K.S.; Jacobsen, K.W. Phonon-limited mobility in n-type single-layer MoS2 from first principles. Phys. Rev. B 2012, 85, 115317. Gamiz, F.; Fischetti, M. V. Monte Carlo simulation of double-gate silicon-on-insulator inversion layers: The role of volume inversion. J. Appl. Phys. 2001, 89, 5478. Poncé, S.; Margine, E.R.; Giustino, F. Towards predictive many-body calculations of phonon-limited carrier mobilities in semiconductors. Phys. Rev. B 2018, 97, 121201.

"

In response to the question about the cut-off, as noted to Reviewer #2: We have tested different kinetic-energy cut-offs and observed that a kinetic energy cut-off of 50 Ry was necessary to get positive and parabolic phonon energies and also symmetric electron(hole)/phonon interaction matrix elements for phonon wave vectors close to the Γ point. We used higher cut-off energies (55 Ry and 60 Ry) during our investigation and no significant change was observed, while a big decrease in the calculation speed was observed. For this reason we used the 50 Ry cut-off in our calculations. As EPW calculations requires the charge density cut-off energy to be 4 times the kinetic energy cut-off, we used a charge density cut-off of 200 Ry.

Round 2

Reviewer 2 Report

'Another consideration we ask the reviewer to make is that if only “high mobility” values can be published, this will strongly skew the publication towards primitive, inaccurate calculations that overestimate mobility.'

I may be naive in this statement, but I strongly believe that the role of a scientist is to provide the community with the most accurate results one can obtain. I would even say that the accuracy is much more important than making a great impression on the readers.

'Regarding the “easily predictable result” we want to make following rebuttal. While it is relatively easy for an expert in the field, familiar with our previous work, to know that scattering with ZA phonons will result in a low mobility, many in community remain unaware of the importance of scattering with ZA phonons.'

I am not sure whether publishing a new and not really impressive paper is the best way to popularize your previous results. If you want to make your paper more recognizable I  would recommend to publish your full-text on the researchgate and put some information about it in your social media (facebook, instagram etc.)

'We make no claims of a new methodology, but rather just a careful application of existing methodology to study the properties of silicane. Nor do we consider how a particular cut-off of ZA phonon scattering could be realized. These could be valuable investigations in their own right but go beyond the scope of our current paper. The analysis we present is entirely within the scope of the journal where many papers are published that do not use a new methodology.'

Neither impressive results nor a new method - it does not sound like a good paper to me. Basing on my experience as a scientist and an author of some papers I know that not all the results can be published in recognizable journals as I believe 'Materials' to be.